# Suicide methods among Brazilian women from 1980 to 2019: Influence of age, period, and cohort

Karina Cardoso Meira[1]*, Raphael Mendonça Guimarães[2], Glauber Weder Santos Silva[3], Rafael Tavares Jomar[4], Eder Samuel Oliveira Dantas[5]

**1** Department of Collective Health at Escola Paulista de Enfermagem, Federal University of São Paulo, São Paulo, Brazil, **2** Sergio Arouca National School of Public Health, Oswaldo Cruz Foundation, Rio de Janeiro, Brazil, **3** Giselda Trigueiro State Hospital, Public Health Secretariat of the State of Rio Grande do Norte, Natal, Brazil, **4** Brazilian National Cancer Institute (INCA), Rio de Janeiro (RJ), Brazil, **5** Onofre Lopes University Hospital, Federal University of Rio Grande do Norte, Natal, Brazil

\* karina.meira@unifesp.br

**Data Availability Statement:** The databases and R codes used in this research are now available at

## Abstract

### Objective

To analyze the effect of age, period, and cohort on suicides among women by hanging, strangulation, suffocation, firearms, and autointoxication in different Brazilian regions from 1980 to 2019.

### Methods

Ecological time-trend study employing estimable functions to estimate APC models, facilitated through the Epi library of the R statistical program, version 4.2.1. Specific rates by age group per 100,00 women and relative risks by period and cohort were estimated using this method.

### Results

Between 1980 and 2019, 49,997 suicides among women were reported using the methods under study. Higher suicide rates per 100,000 women were observed in the South using strangulation and suffocation (2.42), while lower firearm suicide rates were observed in the Northeast (0.13). After adjusting the APC model, there was an increase in age-specific rates with advancing age across all regions for suicides by hanging, strangulation, and suffocation. In contrast, suicides by firearms and autointoxication showed a decrease in rates with advancing age. The period effect indicated an increased risk of suicides by hanging, strangulation (RR >1 and p<0.05) in the five-year intervals of the 2000s in the North, Southeast, and South regions. During the same period, there was an increased risk of suicides by autointoxication in the Southeast, South, and Northeast (RR>1, p<0.05). Suicides by firearms exhibited a statistically significant reduction in the risk of death from 2005 to 2019 in the Southeast and South regions, and from 2005 to 2014 in the Northeast and Midwest. The observed increase in the North region was not statistically significant (RR>1, p>0.05). The

this electronic address. https://zenodo.org/records/13172958.

**Funding:** The study received funding from the National Council for Scientific and Technological Development (CNPQ-306652/2022-6).

**Competing interests:** The authors have declared that no competing interests exist.

cohort effect demonstrated an increased risk of suicides by hanging, strangulation in younger cohorts (RR>1, p<0.05), whereas other methods showed an elevated risk in older cohorts relative to the 1950–1954 generation.

## Conclusion

The results presented here may suggest changes in suicide method preferences between 1980 and 2019.

## Introduction

Suicide affects people from different backgrounds, ethnical groups, socio-economic status and geographical locations [1–6]. Suicide can be defined as any deliberated act by which an individual's death results directly or indirectly from a self-inflicted injury or poisoning [1–6]. Considered a sociocultural phenomenon, it is attributed to the interaction of individual and contextual factors [1–6]. At the individual level, noteworthy factors include depression, bipolar affective disorder, schizophrenia, anxiety, prior history of suicide attempts, excessive alcohol and drug use, and philosophical and existential questions [7–11]. Contextual factors include economic and health crises, as well as societies characterized by fragility in social cohesion [7–11].

The fragility in social cohesion has been documented in patriarchal societies with high levels of inequality in gender, race, and class relations [7, 12–16]. In these societies, the illness and death of women and men are influenced by inequalities in gender relations [14–16]. The construct of gender is an inherent dimension of societal life, subject to various definitions over time [15–19]. Joan Scott [17] argues that gender emerges from the relationship of submission and oppression of women by men, shaping the notions of being male and being female in these societies. In other words, it defines what the thoughts, feelings, and behaviors of women and men should be through the ideal notions of masculinity and femininity, which are presented to individuals via gender technologies [6, 10, 13, 19].

The gender technologies are represented by linguistic codes and cultural representations such as cinema, media, games, and toys. These technologies are constantly performed by family, religion, school, art, media, and other institutions [6, 10, 13, 19]. Zanello [19] asserts that men's subjectivity is configured by gender technologies that valorize the devices of sexual and economic efficacy, while women's subjectivity is shaped by the amorous and maternal devices. Thus, suicidal behaviors in men and women would be aligned with culturally defined gender scripts [20–23].

The selection of methods for attempted suicide by men and women, it is not a random however it is influenced by factors such as accessibility, familiarity, cultural, religious, and socioeconomic issues, as well as the performance of hegemonic masculinity and femininity [20–22, 24]. This dynamic contributes to men opting for more violent and lethal methods to ensure that their masculinity and virility are not questioned in the face of a perceived survival of a suicide attempt. Conversely, women tend to choose less violent and lethal methods as these are socially more acceptable for them [6, 13, 20–22, 24].

Behaviors indicative of the gendered nature of suicidal behavior (ideation, attempts, and death by suicide) [20–24] contribute to the phenomenon known as the suicide paradox or gender paradox. This concept refers to the fact that men are more likely to die by suicide, whereas

women exhibit higher rates of suicidal ideation and attempts, and thus are more broadly affected by suicidal behavior [20–22].

In this context, gender-based violence emerges as a significant factor related to suicidal behavior in women. Devries et al. (2011) argue that suicidal behavior in women can be explained by the interaction between sociodemographic factors (including age, marital status, migration, and socioeconomic status), inequities or lack of access to mental health services, history of childhood violence, experiences of violence in adulthood, and the presence of gender norms in society (such as polygamy, dowry, and family-arranged marriages, among other practices) [12].

Brazil is a country with high gender inequality; women earn only 76% of the average income of men, despite having higher levels of education. They also have limited representation in politics and in managerial positions in public and private enterprises [24]. Furthermore, the country is considered one of the most violent for women, maintaining high rates of physical, sexual, and psychological violence, as well as femicides and female homicides [24, 25].

Violence against women in Brazil is a historical problem that persists despite the implementation of legislation in the 2000s aimed at punishing offenders and protecting women in situations of violence. This includes the stricter implementation of the Maria da Penha Law, the femicide law, and the National Policy for the Reduction of Morbidity and Mortality from Accidents and Violence, which have guided the health sector's response to these issues [26–28].

However, the implementation of these laws has not been matched by an increase in the budget for building a network of protection and support services for women in situations of violence [24, 28]. Consequently, more than a decade after the implementation of these policies, Brazil still has an insufficient number of specialized services to assist women in situations of sexual violence, shelter homes, specialized mental health care services for victims, and specialized police stations for women in situations of violence [24].

The high rates of violence against women, coupled with the inefficiency of the state in preventing, protecting, and supporting women in situations of violence, may be correlated with the increased rates of suicide attempts and suicides observed among Brazilian women [12, 29, 30]. Brazilian studies have shown an increase in suicide rates among women across all age groups, with a heightened risk of mortality in younger generations [31–33]. There has also been an observed shift towards the use of more lethal methods, such as hanging/strangulation/suffocation, and a decline in the use of firearms and autointoxication (with pesticides, insecticides, and medications) [33, 34].

Understanding the most commonly used methods of suicide is an important tool for developing strategies to prevent future deaths. The WHO document (LIVE LIFE: An Implementation Guide for Suicide Prevention in Countries) asserts that effective suicide prevention measures necessarily involve restricting access to means of suicide [34]. This is only feasible when the behavior of the population in choosing methods over time is well understood. The temporal evolution of suicides among women may exhibit significant differences according to age group, study period, generation, and means of suicide [31–33]. Therefore, in studies on the temporal trends of suicides by method, it is important to analyze the effects of three temporal factors (age, period, and cohort) on the behavior of the time series [30–33, 35].

The age effect represents changes in the risk of illness and death associated with different age groups, which may arise due to physiological changes, accumulation of social experience, social roles, changes in social status, or a combination of these factors. The period effect involves structural transformations that affect all age groups simultaneously, such as major sociocultural, economic, and environmental changes (major wars, pandemics, economic crises, health policies, policies restricting access to means of suicide, and policies expanding access to mental health services, among others). The cohort effect manifests as members of

different cohorts experience the impact of these sociocultural, economic, and environmental changes differently throughout their lives. Therefore, exposure to risk and protective factors in the development of diseases and health conditions varies according to age group [33, 35, 36].

Considering the importance of age, period, and cohort effects on the temporal evolution of suicide method choices [33, 34], the objective of this study is to analyze the effects of age, period, and cohort on suicides according to the most commonly used methods in Brazil, specifically hanging, strangulation/suffocation (HSS), firearms (FA), and autointoxication (AUT), across Brazilian regions from 1980 to 2019.

## Materials and methods

### Study design

An observational study with an ecological time-trend design was conducted to analyze the effect of age, period, and cohort on suicides among women by hanging, strangulation/suffocation, autointoxication, and firearm from 1980 to 2019 in Brazilian regions. This study followed the recommendations of the Guidelines for Accurate and Transparent Health Estimates Reporting (GATHER) statement, describing the data sources (death records and population), selected variables, calculated indicators, method of analysis, and the description and discussion of the results [37], as detailed in S1 Table.

### Sociodemographic indicators and gender inequality indicators in study population

Brazil is divided into 26 states and a Federal District. These states are further grouped into five major geographic regions: North, Northeast, Southeast, South, and Midwest. As of 2022, the estimated population of Brazil reached 203.1 million inhabitants. The Southeast region boasts the largest population contingent, closely followed by the Northeast, while the North and Midwest regions have the smallest population quantities. These five geographic regions in Brazil exhibit substantial socioeconomic disparities [38].

The North region, characterized by low population density, encompasses a vast territorial extension, including a significant portion of the Amazon rainforest. The Southeast region, recognized as the most populous, is distinguished by its robust job market. On the other hand, the Midwest, despite being home to the country's capital, focuses its economy primarily on agriculture and livestock (Table 1) [38]. The population residing in the South and Southeast regions exhibits better indicators of income and education, whereas the North and Northeast regions display the poorest indicators in these aspects. Regarding connection to the sewage system, worse access to sanitation was observed in the North (24.41%) and Northeast (43.06%) regions [38] (Table 1). Concerning the income ratio between men and women, men had higher incomes than women in all regions, with greater income inequality occurring in the South (0.73) and Southeast (0.74) regions, where average income rates for both men and women were higher. Conversely, in regions where average incomes were lower, we observed less inequality in income between men and women [25, 38] (Table 1).

### Data sources and study variables

The data used in this study were freely accessed from the Mortality Information System of the SUS Department of Informatics (SIM/DATASUS) [34]. This system is universally accessible. SIM/DATASUS is the information system of the Brazilian Ministry of Health, which provides death records for all Brazilian states and municipalities since 1979. In the present study, the microdata for each Brazilian region was collected annually from 1980 to 2019.

**Table 1. Sociodemographic indicators and gender inequality indicators in Brazil and its major geographic regions, selected years (2018–2022).**

| Region | Variables | | |
|---|---|---|---|
| | Population (n)[a] | 60 or older (%)[b] | 0 to 9 years old[c] |
| North | 17.354.884 | 5.36 | 8.17 |
| Northeast | 54.658.515 | 8.05 | 6.75 |
| Southeast | 84.840.113 | 9.98 | 5.83 |
| South | 29.937.706 | 9.77 | 6.04 |
| Midwest | 16.289.538 | 6.87 | 7.08 |
| Region | Population density (inhabitants per km$^2$)[d] | Connected to the sewage system (%)[e] | Illiterate (%)[f] |
| North | 4.51 | 24.41 | 8.20 |
| Northeast | 35.21 | 43.06 | 14.20 |
| Southeast | 91.70 | 86.68 | 3.90 |
| South | 51.91 | 67.73 | 3.50 |
| Midwest | 10.14 | 54.27 | 5.10 |
| Region | Literacy rate, female (male) (%)[g] | Annual Household Income per Capita USD ($)[h] | Female/male earnings ratio[i] |
| North | 93.0 (94.6) | 2,771.60 | 0.93 |
| Northeast | 87.5 (84.6) | 2,570.50 | 0.87 |
| Southeast | 96.6 (96.9) | 5,084.50 | 0.74 |
| South | 96.4 (97.0) | 4,993.30 | 0.73 |
| Midwest | 95.2 (95.1) | 4,672.80 | 0.76 |

Note: [a]Total population according to the 2022 Census (https://censo2022.ibge.gov.br/panorama/); [b]Percentage of the female population aged 60 or over according to the 2022 Census (https://censo2022.ibge.gov.br/panorama/);[c]Percentage of the female population aged 0 to 9 according to the 2022 Census (https://censo2022.ibge.gov.br/panorama/);[d]Population density (inhabitants per km$^2$) according to the 2022 Census(https://censo2022.ibge.gov.br/panorama/); [e]Percentage of the Connected to the sewage system according to the 2022 Census(https://censo2022.ibge.gov.br/panorama/);[f] Percentage of people aged 15 or over who are illiterate according to the 2022 Census(https://censo2022.ibge.gov.br/panorama/);[g]Adult literacy rate (female and male), being the percentage of people aged 15 and above who can both read and write with understanding a short simple statement about their everyday life (BRAZIL, 2021); [h] Monthly Household Income per Capita (BRAZIL, 2018);[i] Female-to-male earnings ratio based on average earnings (BRAZIL, 2021)

The microdata is available in dbc format and was converted to dbf format using the Tabwin program version 4.15 for Windows provided by the Brazilian Ministry of Health. After converting the data to dbf format, the death records for each year (1980 to 2019) were aggregated for each of the Brazilian regions using the R software (version 4.1), extracting only death records of females aged 10 years and older. Death records were extracted from the Mortality Information System of the Department of Informatics of the Unified Health System (SIM/DATASUS) [34], considering the 9th and 10th editions of the International Classification of Diseases and Related Health Problems (ICD-9 and ICD-10), as per the codifications presented in Table 2.

Population data for mortality rate estimates were also obtained from DATASUS in the sociodemographic and economic data section, based on a demographic census from 1980, 1991, 2000 and 2010. The Brazilian Institute of Geography and Statistics estimated populations projections on July 10 of the intercensal years [38].

We analyzed all death records classified as suicides in women aged 10 and older, utilizing the methods of hanging/strangulation/suffocation, firearm, or autointoxication. These incidents occurred from 1980 to 2019 across the five major regions of the country.

The variables collected pertained to geographic region (North, Northeast, Southeast, South, and Midwest), age group (10 to 14, 15 to 19, 20 to 24, 25 to 29, 30 to 34, 35 to 39, 40 to 44, 45 to 49, 50 to 54, 55 to 59, 60 to 64, 65 to 69, 70 to 74, 75 to 79, and 80 years or older), year of death

**Table 2. Death records sourced from the Mortality Information System of the SUS information technology department, categorized by ICD-9 and ICD-10.**

| Health problem | ICD-9 | ICD-10 |
|---|---|---|
| **Suicide** | | |
| Suicide by firearm | E955 | X72 to X74 |
| **Other external causes of accidental injuries** | | |
| Other external causes of accidental injuries by hanging, strangulation, and suffocation. | E913 | W75 to W76 |
| Other external causes of accidental injuries by accidental autointoxication. | E850 to E869 | X40 to X49 |
| Other external causes of accidental injuries by firearm | E922 | W32 to W34 |
| **Assault** | | |
| Homicide by hanging/strangulation/suffocation | E963 | X91 |
| Homicide by autointoxication | E962 | X85 to X90 |
| Homicide by firearm | E965 | X93 to X95 |
| **Event of undetermined intent** | | |
| Hanging/strangulation/suffocation, intent undetermined | E983 | Y20 |
| Intoxication or autointoxication, intent unknown (whether intentional or accidental) | E980 to E982 | Y1 to Y19 |
| Firearm discharge, rifle, pistol, other firearms, intent undetermined | E985 | Y22 to Y24 |
| Sequelae of self-inflicted injury | E959 | Y87 |

Source: 9th and 10th editions of the International Classification of Diseases and Related Health Problems.

(1980 to 2019), and health problem (Suicide, Other external causes of accidental injuries, Assault, and Event of undetermined intent).

Age groups from 10–14 years to 80 years or older were chosen due to an excess of zeros in smaller age groups. We opted to include the 10–19-year age group because, in Brazil, an increase in suicide rates has been observed in this demographic over the last decade [29, 31–33]. Additionally, Brazilian children and adolescents in this age range, particularly in regions with greater socioeconomic vulnerability, are already exposed to complex situations such as sexual violence, child marriage, and teenage pregnancy, which may influence their risk of suicide [15, 19, 39, 40].

## Correction for poor certification and underreporting of death records

The Mortality Information System (SIM) displays varying quality in death certification, coverage, and completeness of variables across different Brazilian regions. Notably, there is a high proportion of death records with undetermined underlying causes and significant underreporting, particularly in the North and Northeast regions [41–44]. In response, we chose to rectify death records for both poor certification and underreporting. Since there is no consensus in the literature on the best method to correct for the poor certification of suicides [30, 31, 33], we adapted the approach proposed by Garcia et al. [45] in six steps, as outlined in S1 Table. The results of the stages of the death record correction process are presented in S2 Table.

After correcting for poorly certified deaths, we addressed underreporting by utilizing the degree of coverage of death records for women, segmented by geographic region and decade. This the degree of coverage of death was estimated using the Adjusted Synthetic Extinct Generations (SEG-adj) death distribution method, which combines the extinct generations and general balancing equation methods [44, 46]. The estimated coverage degree was provided by researchers from the Population Estimates and Projections Laboratory at the Federal University of Rio Grande do Norte [47]. The rectification process of the records was conducted by two independent researchers and verified by a third.

## Statistical analyses

**Exploratory analyses.** Following the correction of death records, we calculated specific mortality rates by age group, both crude and standardized using the direct method, with the World Health Organization's proposed global standard population. Mortality coefficients per 100,000 women were computed based on the means (firearm, autointoxication, and hanging/strangulation/suffocation). To assess the effect of age, period, and cohort, we grouped age ranges and periods into five-year intervals: age group (10 to 14, 15 to 19, 20 to 24, 25 to 29, 30 to 34, 35 to 39, 40 to 44, 45 to 49, 50 to 54, 55 to 59, 60 to 64, 65 to 69, 70 to 74, 75 to 79, and 80 years or older), period (1980 to 1984,1985 to 1989,1990 to 1994, 1995 to 1999, 2000 to 2004, 2005 to 2009, 2010 to 2014 and 2015 to 2019), and cohort (1900 to 1904, 1905 to 1909, 1910 to 1914, 1910 to 1914, 1915 to 1919, 1920 to 1924, 1925 to 1929, 1930 to 1934, 1935 to 1934, 1940 to 1944, 1945 to 1949, 1950 to 1954, 1955 to 1959, 1960 to 1964, 1965 to 1969, 1970 to 1974, 1975 to1979, 1980 to 1984, 1985 to 1989, 1990 to 1994, 1995 to 1999, 2000 to 2004 and 2005 to 2009).

In the descriptive analysis of the temporal evolution of mortality rates by year and method, we smoothed the corrected suicide mortality coefficients using triennial moving averages, categorized by means (hanging/strangulation/suffocation, firearm, and autointoxication). After smoothing the rates, line graphs were constructed to assess the temporal evolution of mortality rates [42].

**Age, period and cohort model.** The APC effects were estimated using regression models with a Poisson distribution for the number of observed deaths in each age group i and period j ($\theta_{ij}$). These effects are additively related to the logarithm of the expected mortality rate (E($r_{ij}$)), following Holford's proposal [35].

$$ln\left(E[r_{ij}]\right) = ln\left(\frac{\theta_{ij}}{N_{ij}}\right) = \mu + \alpha_i + \beta_j + \gamma_k$$

Where $E[r_{ij}]$ denotes the expected rate, $\theta_{ij}$ the number of observed deaths, and $N_{ij}$ the population at risk of death in age group $i$ and period $j$. The parameter $\mu$ represents the average effect, $\alpha_i$ represents the effect of age group $i$, $\beta_{ij}$ the effect of period $j$, and $\gamma_k$ the effect of cohort $k$ (k = 1,...K; K = I + J − 1 = 22). Here, i = 1, ..., I; j = 1, ... J; k = 1, ..., K. Where I correspond to the number of age groups, J to the number of periods, and K to the number of cohorts. Consequently, in this study we obtained I = 15 age groups, J = 8 periods, and K = I + J− 1 = 22 birth cohorts (1900 to 2009).

The estimation of APC effects has a primary limitation known as the non-identifiability problem of parameters in the complete model due to the linear relationship among temporal effects (I = J—K). Various methodological tools have been proposed to address this limitation, but there is no consensus in the literature on the best methodology. In this study, APC effects were estimated using estimable functions, as proposed by Holford [35, 36] and implemented in models adjusted by the Epi library (https://CRAN.R-project.org/package=Epi) in the R software (https://www.R-project.org/) [48].

Estimable functions are limited to the analysis of linear combinations and curvature effects of temporal terms (age, period, and cohort). The linear trend of effects is divided into two components: the linear effect of age and the drift effect (linear effect of period and cohort). The longitudinal trend of age is the sum of age and period slope ($\alpha L + \beta L$), where $\alpha L$ and $\beta L$ are the linear trends of age and period, respectively. The second drift term represents the linear trend of the logarithm of specific rates (mortality) by age and is equal to the sum of period and cohort slopes ($\beta L + \gamma L$), where $\beta L$ and $\gamma L$ are the linear trends of period and cohort, respectively [35, 48, 49].

Fifteen APC analyses (scenarios) were conducted for female suicides by firearm, autointoxication, and hanging,strangulation/suffocation in the five Brazilian regions. In each scenario, the adjusted APC submodels were compared in a nested manner via deviance statistics and likelihood ratio tests at a 5% significance level, as proposed by Holford [35, 49].

The deviance analysis of the APC models determined by the Epi library estimates six nested equations, namely: (1) f(a) age; (2) f(a) + δc age-drift; (3) f(a) + h(c) age-cohort; (4) f(a) + g(p) + h(c) age-period-cohort; (5) f(a) + g(p) age-period; and (6) f(a) + δp age-drift. Here, a, p, and c represent the effects of age, period, and cohort; f, h, and g are smooth functions of parameters; and δ is a linear effect [35, 48].

Based on the best-fit model, estimated specific mortality rates by age, and relative risks (RR) were extracted for each period and cohort concerning their respective reference categories (period: 2000–2004; birth cohort: 1950–1954). Interval estimates were obtained at a 95% confidence level [35, 49]. We chose the five-year period 2000–2004 as it marks the implementation of significant suicide prevention measures (National Mental Health Policy and Disarmament Statute and National Policy for the Reduction of Morbidity and Mortality from Accidents and Violence) [27, 33, 50, 51]. Regarding the reference cohort, we selected 1950–1954 because median cohorts tend to have a greater quantity of values, being more stable and complete than the first and last ones [35, 36, 48, 49]. Furthermore, previous studies conducted in Brazil have shown a lower risk of death in generations born from the 1950s onwards [33, 50, 51].

## Ethical considerations

The data used are in the public domain and do not identify individuals (https://datasus.saude.gov.br/mortalidade-desde-1996-pela-cid-10) [25]. Therefore, the study was exempt from review by an Ethics Research Committee.

## Results

### Exploratory analyses

From 1980 to 2019, Brazil reported 49,997 suicides by hanging, strangulation/suffocation (55.73%), autointoxication (32.71%), and firearms (11.57%). After adjusting for misclassification and underreporting of death records, there was a 34.62% increase in recorded suicides by the studied methods. The highest percentage increase in suicide rates was observed during the 1980s and 1990s, particularly in the North and Northeast regions (S2 Table).

The highest suicide rates by hanging, strangulation, and suffocation (HSS) and firearm (FA) were observed in the South region, 2.42 and 0.60 per 100,000 women, respectively. Suicides by autointoxication (AUT) exhibited higher mortality rates in the Midwest (1.52 suicides per 100,000 women). In the Southeast region, we observed the lowest rates of suicide by HSS and AUT, respectively, 0.85 and 0.77 per 100,000 women, while the Northeast showed the lowest suicide rates by FA (0.13 suicides per 100,000 women) (Table 3).

**Age effect.** Suicide rates by hanging, strangulation, and suffocation in women, according to age groups, show differences among the Brazilian regions. In the Northeast and Southeast, there was a positive gradient in the magnitude of the rates, with a peak incidence in the 50–54 age group. The South region exhibited a peak incidence among elderly women (70–74 years). In the North and Midwest regions, there was a negative gradient in the magnitude of suicide rates with increasing age, with the highest rates observed among adolescent women (15 to 19 years) (Fig 1).

The suicide rates by autointoxication in the North, Northeast, and Midwest regions were higher among adolescents aged 15 to 19 years, with a progressive decline as age increased. In the South and Southeast regions, there was a progressive increase in suicide rates up to the

**Table 3. Suicide rates per 100,000 women, by perpetration method, region, and quinquennium.** Brazil, 1980–2019.

| Perpetration method | Period | Locality | | | | |
|---|---|---|---|---|---|---|
| | | North | Northeast | Southeast | South | Midwest |
| Hanging/strangulation/suffocation | 1980 to 1984 | 1.02 | 0.50 | 0.59 | 2.81 | 0.47 |
| | 1985 to 1989 | 0.79 | 0.58 | 0.59 | 2.82 | 0.91 |
| | 1990 to 1994 | 0.87 | 0.68 | 0.66 | 2.59 | 0.75 |
| | 1995 to 1999 | 0.84 | 0.71 | 0.70 | 2.39 | 1.24 |
| | 2000 to 2004 | 1.51 | 0.89 | 0.62 | 2.10 | 1.63 |
| | 2005 to 2009 | 1.62 | 1.32 | 0.78 | 2.11 | 1.69 |
| | 2010 to 2014 | 2.05 | 1.43 | 0.99 | 2.38 | 1.75 |
| | 2015 to 2019 | 2.96 | 1.87 | 1.51 | 2.76 | 2.52 |
| | SAR | 1.22 | 1.11 | 0.85 | 2.42 | 1.53 |
| Autointoxication | 1980 to 1984 | 1.10 | 0.54 | 1.20 | 2.02 | 2.43 |
| | 1985 to 1989 | 0.95 | 0.37 | 0.87 | 1.38 | 2.30 |
| | 1990 to 1994 | 0.67 | 0.44 | 0.73 | 1.03 | 1.50 |
| | 1995 to 1999 | 1.21 | 0.46 | 0.65 | 0.85 | 1.48 |
| | 2000 to 2004 | 1.05 | 0.89 | 0.81 | 0.79 | 1.66 |
| | 2005 to 2009 | 0.78 | 1.30 | 0.94 | 0.85 | 1.27 |
| | 2010 to 2014 | 0.62 | 1.17 | 0.95 | 0.87 | 1.30 |
| | 2015 to 2019 | 0.56 | 0.93 | 0.73 | 1.05 | 1.07 |
| | SAR | 0.98 | 0.84 | 0.77 | 1.08 | 1.52 |
| Firearm | 1980 to 1984 | 0.31 | 0.21 | 0.35 | 0.86 | 0.66 |
| | 1985 to 1989 | 0.43 | 0.16 | 0.32 | 0.72 | 0.66 |
| | 1990 to 1994 | 0.36 | 0.22 | 0.34 | 0.89 | 0.71 |
| | 1995 to 1999 | 0.27 | 0.19 | 0.38 | 0.98 | 0.70 |
| | 2000 to 2004 | 0.18 | 0.13 | 0.33 | 0.64 | 0.48 |
| | 2005 to 2009 | 0.14 | 0.09 | 0.24 | 0.43 | 0.29 |
| | 2010 to 2014 | 0.12 | 0.07 | 0.19 | 0.31 | 0.16 |
| | 2015 to 2019 | 0.09 | 0.07 | 0.17 | 0.23 | 0.22 |
| | SAR | 0.20 | 0.13 | 0.28 | 0.60 | 0.38 |

Note: SAR = standardized mortality rate

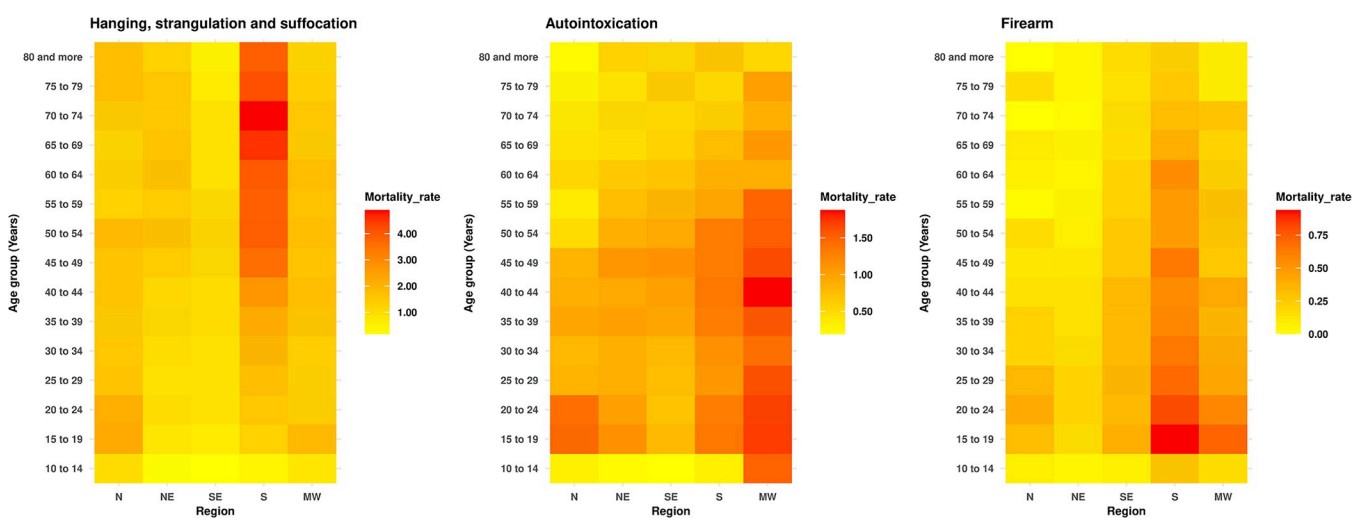

**Fig 1. Suicide rates per 100,000 women, by perpetration method, age group, and region.** Brazil, 1980–2019.

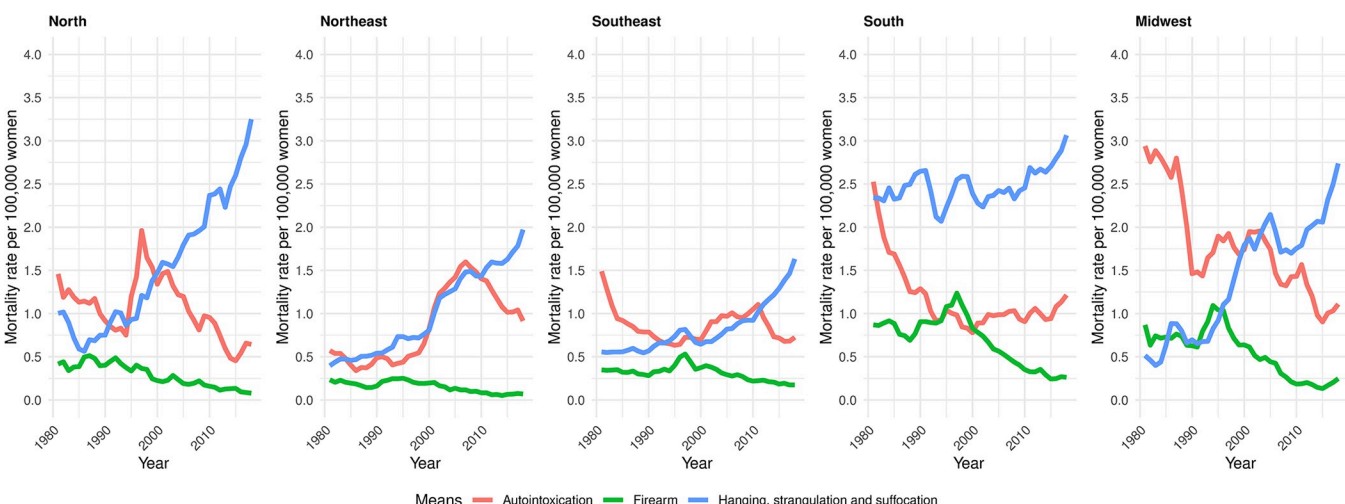

**Fig 2. Suicide rates per 100,000 women, smoothed by triennial moving averages, according to perpetration method and region.** Brazil, 1980–2019.

fourth decade of life (Fig 1). In all locations, we observed higher rates of firearm suicide among women in their second decade of life, followed by a progressive decline with advancing age (Fig 1).

**Period effect.** Over the forty-year study period, we observed a progressive increase in suicide rates by hanging, strangulation, and suffocation beginning in the mid-2000s in the North, Southeast, and Midwest regions, and from the mid-1990s in the South region (Fig 2 and Table 3). In contrast, suicide rates by firearm exhibited a declining trend across all regions starting in the late 1990s (Fig 2 and Table 3).

Suicides due to autointoxication also exhibited a temporal trend of decreasing mortality rates across all Brazilian geographic regions. In the North and Central-West regions, a reduction in suicide rates was observed beginning in the early 2000s. In the Northeast and Southeast, the decline in rates began from 2009 onwards. In the South, a reduction in suicide rates due to self-intoxication was noted from the 1990s onward, with a subtle increase observed in the final years of the historical series (Fig 2 and Table 3).

**Cohort effects.** Suicide rates by cohort and age group for suicides involving hanging, strangulation indicate an increase in rates among women born since the 1950s who are between 50 and 54 years old (Fig 3). Conversely, for suicides due to autointoxication, there was a reduction in rates starting with the 1940–1944 cohort (age group 60 to 64 years) across all regions except the Northeast, where the decrease was observed in the 1965–1969 cohort (age group 40 to 44 years). In the North region, there was an increase in rates among women from the 1940–1944 cohort (ages 60 to 64) (Fig 4). Regarding suicides by firearm, we observed a progressive reduction in rates among women from the 1930–1934 cohort who are between 50 and 54 years old (Fig 5).

**Age-period-cohort models.** The analysis of AIC and deviance indicated that the full model (age, period and cohort) provided a better fit to the data in all regions and means of perpetration (S3 Table).

**Age period.** The age-specific rates adjusted for period and cohort effects for suicides by hanging, strangulation, and suffocation exhibited a progressive increase up to the age group of 50–54 years in the South and Southeast regions. In contrast, in other regions, the peak incidence occurred among women aged 80 years or older (Fig 6). Suicides by autointoxication and firearm displayed an opposite pattern, with a progressive decrease in rates with advancing age,

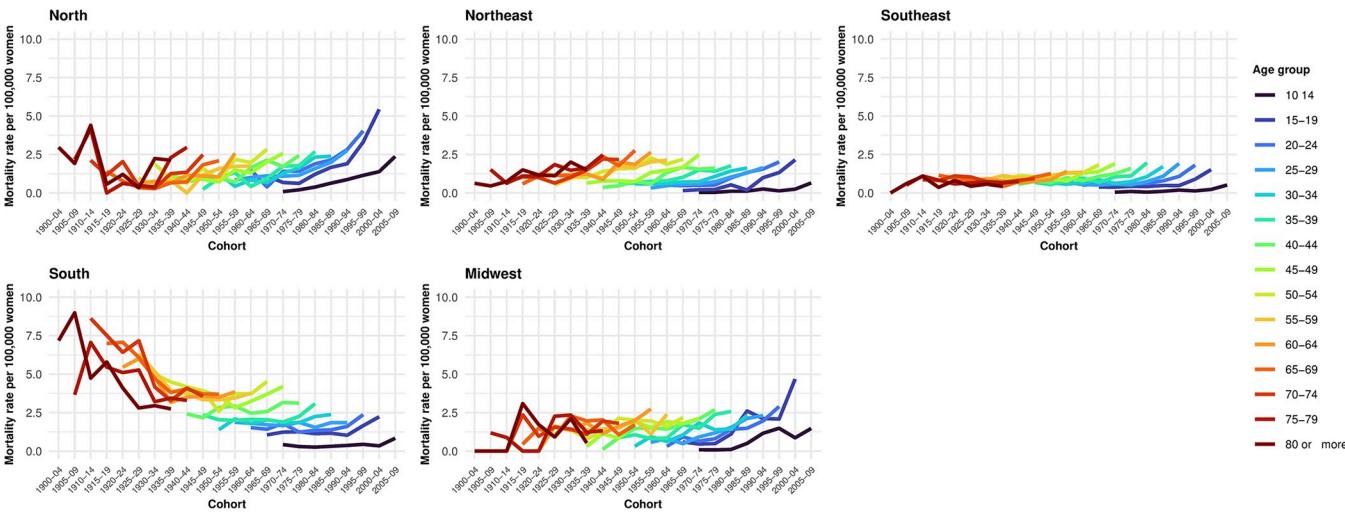

**Fig 3. Suicide rates by hanging/strangulation/suffocation, according to cohort, age, and region.** Brazil, 1980–2019.

except for suicides by autointoxication in the Northeast, which showed an increase with age (Fig 6).

## Period effect

After adjusting the Age-Period-Cohort (APC) models using the estimable functions method, the trend over the 40-year study period, as measured by the age-drift linear trend, indicated an upward trajectory for suicides by hanging, strangulation, and suffocation in all Brazilian regions except the South, where the trend was stationary. Suicides by autointoxication and firearms exhibited a downward trend in all regions, except for autointoxication suicides in the Northeast, which showed an upward trend (Table 4).

The relative risk by period adjusted for the effect of age and cohort (RR), using the period from 2000 to 2004 as a reference, showed an increase in the risk of death (RR>1, p<0.05) in the five-year periods of the 2000s in the North, Southeast, and South regions. In the Midwest

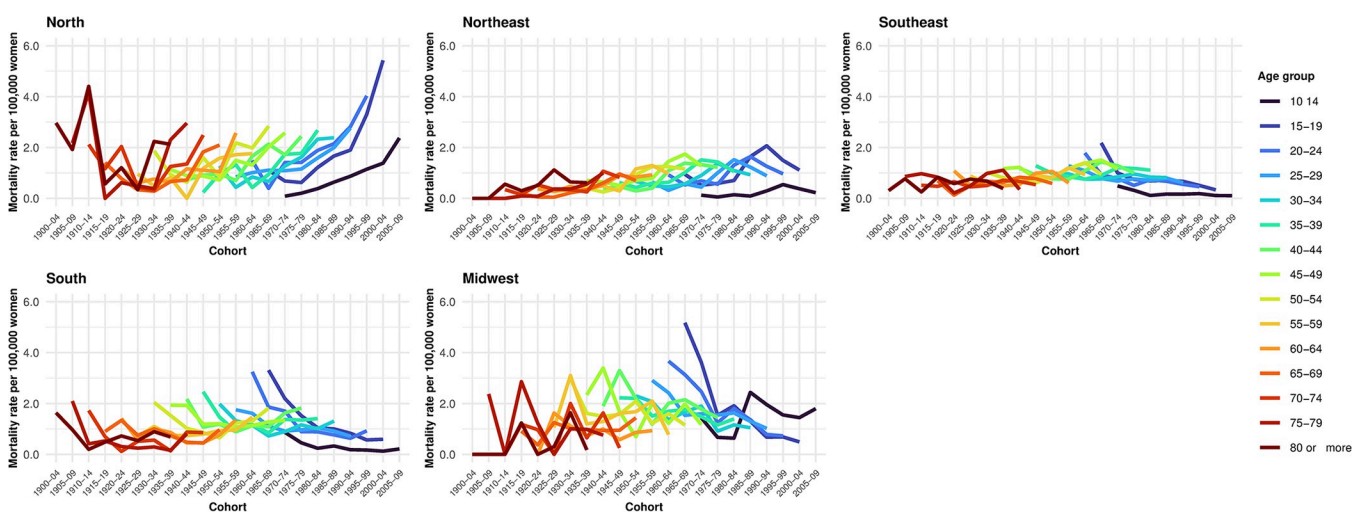

**Fig 4. Suicide rates by autointoxication, according to cohort, age, and region.** Brazil, 1980–2019.

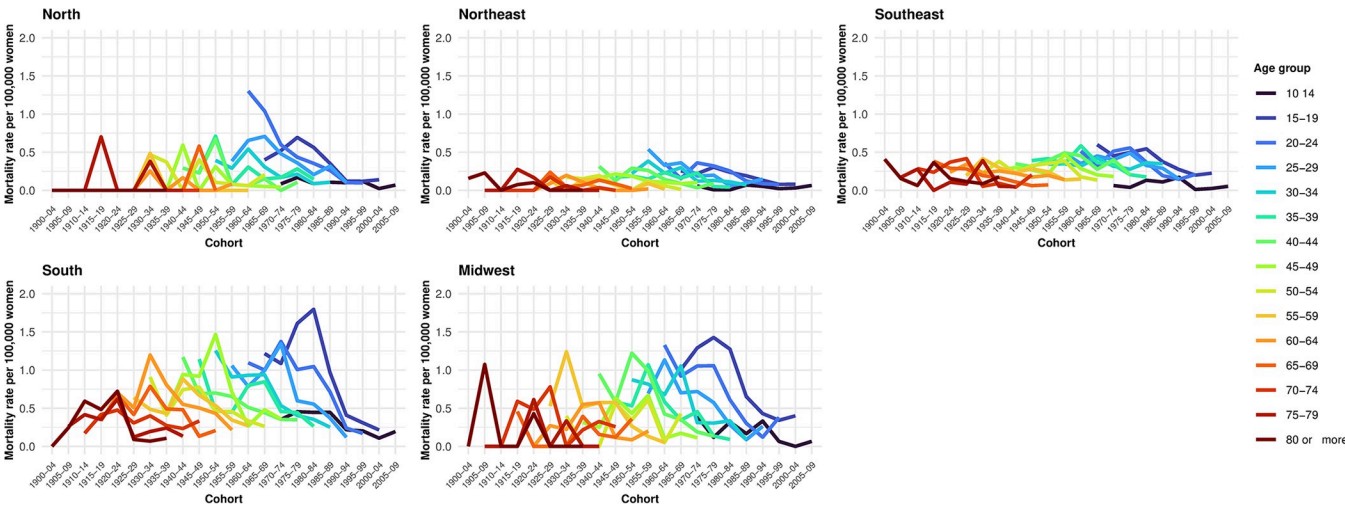

**Fig 5. Suicide rates by firearm, according to cohort, age, and region.** Brazil, 1980–2019.

region, there was a reduction in the risk of death (RR<1, p<0.05), while in the Northeast, the reduction was not statistically significant at the 5% level (RR<1, p>0.05) (Fig 7 and S4 Table).

For suicides by autointoxication in the five-year periods of the 2000s, there was a statistically significant increase in the risk of death (RR>1, p<0.05) in the Southeast, South, and Northeast regions during the periods 2005–2009 and 2010–2014. In the North, there was a reduction in risk during the period of 2005 to 2019, and from 2015 to 2019 in the Northeast. In the Midwest, the decrease was not statistically significant (RR<1,p>0.05) (Fig 7 and S4 Table).

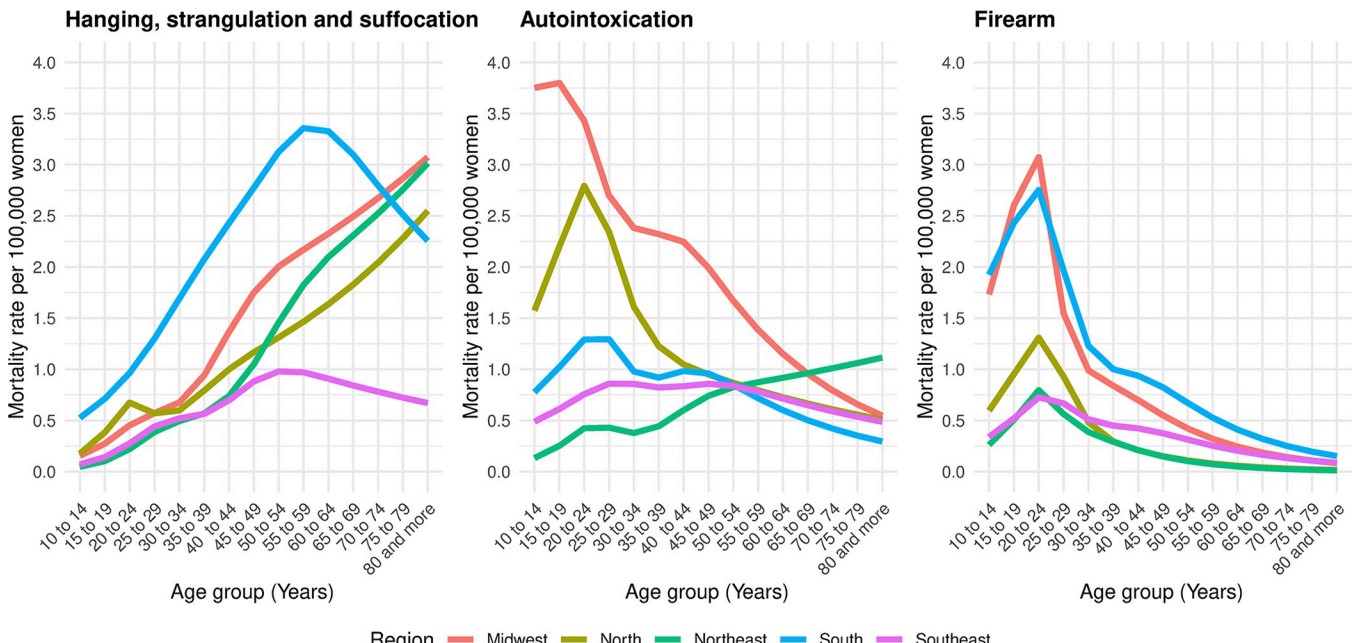

**Fig 6. Suicide rates per 100,000 women adjusted for the period and cohort effect, by age group, method of perpetration, and region.** Brazil, 1980–2019.

**Table 4. Linear age-drift trend for female suicides, by means of perpetration and region.** Brazil, 1980–2019.

| Locality | Perpetration method | Age-drift (CI-95%) | Trend |
|---|---|---|---|
| **North** | Hanging/strangulation/suffocation | 1.030 (1.020–1.040) | Upward |
| | Autointoxication | 0.980 (0.975–0.984) | Downward |
| | Firearm | 0.963 (0.948–0.971) | Downward |
| **Northeast** | Hanging/strangulation/suffocation | 1.037 (1.034–1.040) | Upward |
| | Autointoxication | 1.029 (1.027–1.032) | Upward |
| | Firearm | 0.964 (0.959–0.970) | Downward |
| **Southeast** | Hanging/strangulation/suffocation | 1.029 (1.027–1.031) | Upward |
| | Autointoxication | 0.992 (0.990–0.994) | Downward |
| | Firearm | 0.980 (0.977–0.983) | Downward |
| **South** | Hanging/strangulation/suffocation | 0.998 (0.996–1.001) | Stationary |
| | Autointoxication | 0.981 (0.979–0.984) | Downward |
| | Firearm | 0.966 (0.963–0.970) | Downward |
| **Midwest** | Hanging/strangulation/suffocation | 1.042 (1.030–1.046) | Upward |
| | Autointoxication | 0.976 (0.973–0.979) | Downward |
| | Firearm | 0.957 (0.951–0.963) | Downward |

Suicides by firearms showed a statistically significant reduction in the risk of death in the period of 2005 to 2019 in the Southeast and South regions, and from 2005 to 2014 in the Northeast and Midwest. The observed increase in the North region was not statistically significant (RR>1, p>0.05) (Fig 7 and S4 Table).

**Cohort effect.** In the North, Southeast, and Midwest regions, a protective effect on the risk of suicide by hanging, strangulation, and suffocation was observed for older cohorts compared to the 1950–1954 cohort (RR<1, p<0.05). Conversely, there was a progressive increase in the risk of death among women born from 1955–1959 onwards (Fig 8 and S5 Table). In the South region, all cohorts exhibited a higher risk of suicide compared to the 1950–1954 cohort (RR>1, p<0.05) (Fig 8 and S5 Table).

Firearm suicides exhibited an inverse trend compared to suicides by hanging, strangulation, and suffocation, as all regions in Brazil showed a reduction in the risk of death from older

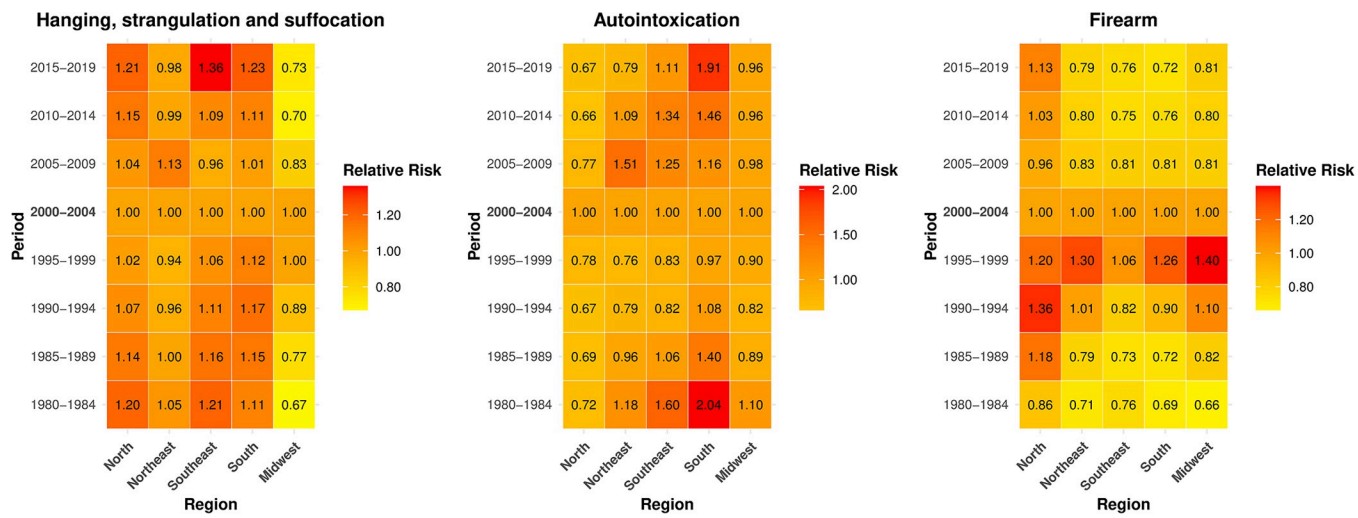

**Fig 7. Risk of suicide in women according to period, adjusted for the age and cohort effect, by method of perpetration, and region.** Brazil, 1980–2019.

## Hanging, strangulation and suffocation

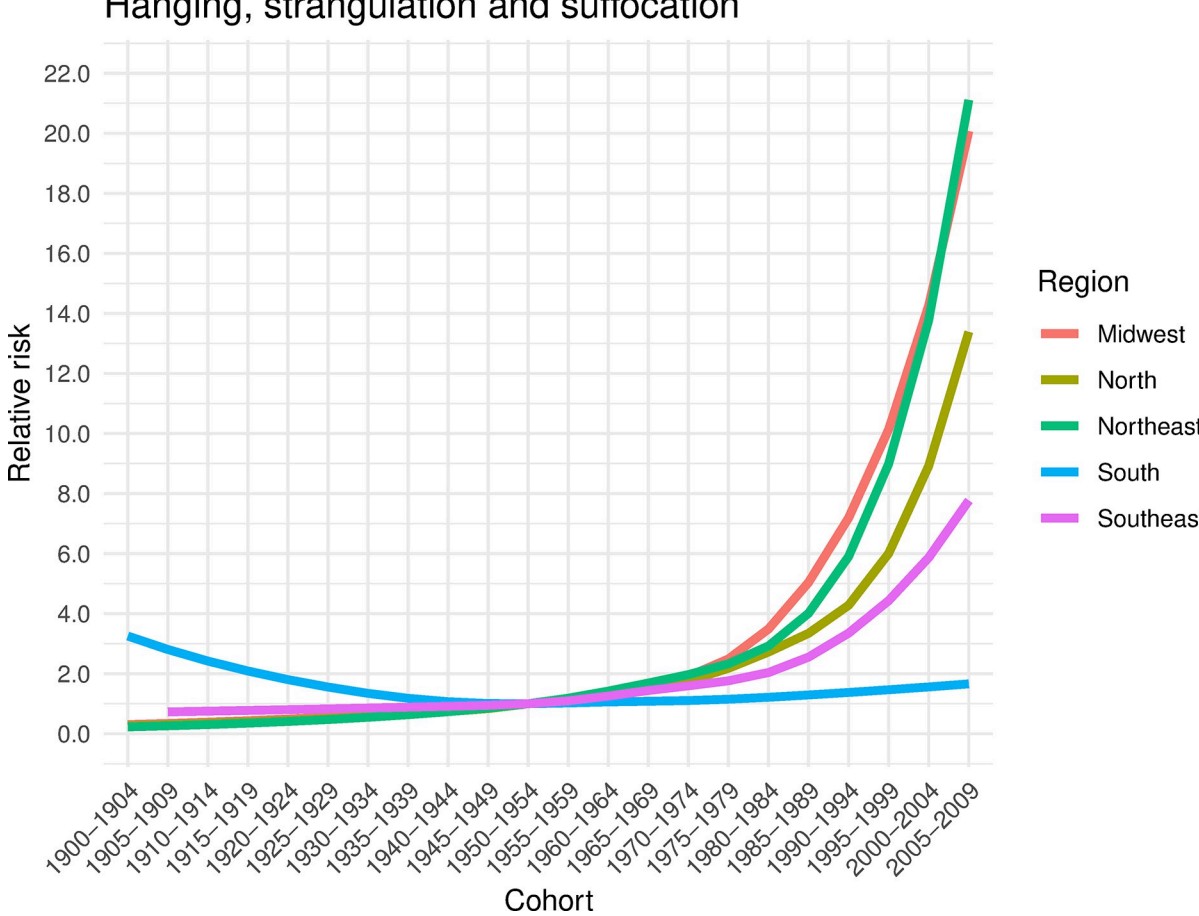

**Fig 8. Risk of suicide by hanging, strangulation and suffocation in women, according to cohort, adjusted for the age and period effect, by region.** Brazil, 1980–2019.

cohorts (1900–1904 to 1945–1949) compared to younger generations (1955–1959 to 2005–2009) (Fig 9 and S4 Table). In suicides by autointoxication, there was also a reduction in the risk of death among younger cohorts (RR<1,p<0.05), with the exception of the Southeast (1955–1979 cohort) and Northeast (1955–2009 cohort), which showed an increased risk among younger women (RR>1,p<0.05) (Fig 9 and S5 Table).

## Discussion

The primary findings of this study identified higher suicide rates among Brazilian women in the South, Midwest, and North regions, with prominent methods being hanging, strangulation/suffocation, and autointoxication. Upon adjustment of the Age-Period-Cohort (APC) models, we observed differences in the effects of age, period, and cohort across these methods.

The age effect indicated a rise in suicide rates with advancing age for suicides by hanging, strangulation/suffocation, and a decrease in suicides by firearm and autointoxication. Moreover, in most Brazilian regions, there was a consistent period effect, characterized by a reduction in the risk of firearm suicides and an increase in the risk of suicides by hanging, strangulation/suffocation, and autointoxication during the five-year intervals from 2005 to 2019, compared to the period from 2000 to 2004.

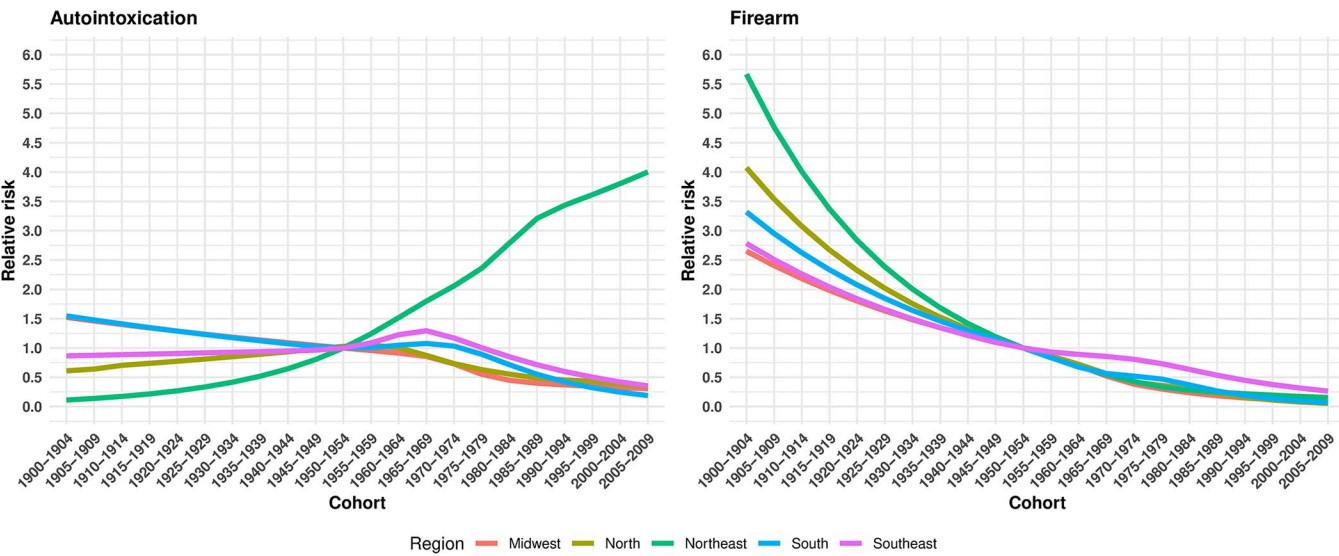

**Fig 9. Risk of suicide in women according to cohort and methods, adjusted for the age and period effect, by region.** Brazil, 1980–2019.

Regarding the cohort effect, an increased risk of suicide by hanging and strangulation/suffocation was observed among younger cohorts (from1955–1959 to 2005–2009) when compared to the reference cohort (1950–1955). Conversely, the risk of suicides by firearm and autointoxication was lower among the younger generations.

The temporal evolution of suicides and the choice of methods for their execution are not random occurrences; rather, they are influenced by social, economic, cultural factors, gender norms, and the availability of methods [6–13, 20–22, 24]. According to Kirmayer (2022), conditions associated with suicidal behavior are consequences of specific lifestyles sustained by values, institutions, and cultural practices [52]. In this context, gender norms and violence prevalent in patriarchal societies significantly influence suicidal behavior among women [12, 13, 17–19, 52, 53].

In Brazil, as in other Latin American countries, society is conservative and patriarchal, establishing distinct social roles, norms, and gender conventions for men and women. Among men, these norms exacerbate a lack of self-care and neglect of both physical and mental health. Such practices not only compromise male health but also lead to negative consequences for women, manifested in interpersonal violence, femicide, and suicides [12–19, 52–56].

For women, suicidal behavior is frequently linked to exposure to violence, abuse, and deprivation in childhood [12, 13, 17–19, 52, 53]. In Brazil, although women exhibit lower suicide rates compared to men, they are more susceptible to suicidal behavior due to a higher prevalence of suicidal ideation and attempts [30, 32, 33, 51, 54, 55, 57]. The temporal trend and magnitude of suicide attempts and suicides among women vary across different Brazilian regions [32, 33, 57, 58, 61].

Historically, higher rates have been observed in the South and Midwest regions; however, in recent years, the North region has also emerged as one of the regions with the highest suicide rates among women [32, 33, 57, 58, 61]. These findings are consistent with our results, which indicate higher suicide rates by hanging, strangulation/suffocation, and autointoxication in these geographic regions of Brazil.

The South region of Brazil has a strong identity formation marked by European colonization, which is accompanied by historically elevated suicide rates [59, 60]. Notably, the Midwest

region experienced a significant migratory influx in the 1970s, with workers coming from the South region, resulting in similar identity formation patterns [58–60]. Additionally, both the South and Midwest share agribusiness as a predominant economic activity [34, 58–60]. This is relevant because rural workers face a high risk of suicidal behavior due to low income, job instability, limited access to healthcare services, and increased exposure to suicide methods such as pesticides and insecticides [58–61].

The socioeconomic and demographic characteristics of the Midwest and North regions add another layer of complexity to the risk of suicide. These regions have a high percentage of Indigenous populations and include border municipalities in the Northern Arc (Amapá, Pará, Amazonas, Roraima, and Acre) and the Central Arc (Rondônia, Mato Grosso, and Mato Grosso do Sul) [25–27, 30, 38, 40]. Historically, these border areas in South America and the Caribbean have been zones of conquest, instability, and violence. The situation is further aggravated by the involvement of these areas in international drug, goods, and human trafficking, making women, children, and Indigenous populations particularly vulnerable to acts of violence [62–64]. These conditions heighten the risk of suicidal behavior among Indigenous people compared to the general population [65–68].

In addition to the regional dynamics that influence the magnitude of suicide rates, our study revealed a significant variation in the effect of age based on the chosen method of suicide. The physiological transformations associated with aging affect the risk of illness and death throughout life, reflecting changes in the epidemiological profile of a community as its age structure evolves (age effect) [36, 37, 49, 69, 70]. This context is particularly relevant because studies have shown that age group plays an important role in the choice of suicide method [70–78].

This age variation in suicide methods is also evident across different Brazilian regions. After adjusting the APC models, we observed that the suicide rates by hanging, strangulation/suffocation progressively increase with age, whereas suicides by autointoxication and firearms display an opposite trend. Similar results have been found in the United States, Lithuania, Ecuador, in the city of Rio de Janeiro (Brazil), as well as in other nations in Latin America and the Caribbean [78–85]. These differences in the age effect in relation to methods may be associated with social, economic, and cultural factors, health issues (both mental and physical), as well as the availability and acceptability of the methods used [6, 10, 12, 13, 19–24].

To understand these variations, it is essential to consider how different factors influence the choice of suicide methods across different age groups. Among elderly women, loneliness and hopelessness, often resulting from social isolation and illnesses, may lead to the selection of methods that minimize the chance of failure in suicide attempts, indicating a strong desire to die. In contrast, among younger individuals, impulsivity often drives the choice of readily available methods, such as insecticides, pesticides, medications, and firearms [20–24, 84, 85].

Given this scenario, it is necessary to discuss the role of socioeconomic, cultural, and health factors in suicidal behavior throughout the life cycle. Among adolescent girls, gender-based violence, teenage pregnancy, and lack of support are significant factors [6, 10, 12, 13, 19, 20–24]. In middle-aged women, suicidal behavior may be influenced by the interaction between gender-based violence and socioeconomic factors, such as unemployment, underemployment, and the physical and mental burden from the double workload, which includes both unpaid domestic work and paid employment [6, 10, 12, 13, 19–24]. For elderly women, suicidal behavior may be related to chronic diseases, disabilities, difficulties in fulfilling the caregiver role, lack of social security, and social isolation [6, 10, 12, 13, 19–24].

Besides the factors specific to each age group, the preference for suicide methods can change over time within the same country [32, 33, 36, 37, 49, 69, 70, 78–85]. These changes may be related to period effects, including restrictions on certain methods, political and

economic contexts, access to health services, and social protection, which impact all age groups [9, 25, 58, 63, 67, 76, 82, 86].

In suicides among women in Brazilian regions from 1980 to 2019, we identified differences in temporal trends based on region and method used. We observed an upward trend in suicides by hanging and strangulation/suffocation, while suicides by autointoxication and firearms showed a downward trend. These results may be related to the substitution of methods, as restrictions on specific means of suicide, such as pesticides/insecticides and firearms, can lead to an increase in deaths by other, more accessible and harder-to-control methods, such as hanging [87–89].

This substitution of methods can be partially explained by the implementation of the 2003 policy (Estatuto do Desarmamento), which restricted the purchase and circulation of firearms within the country. Although this measure contributed to a reduction in suicides by firearms, it was not sufficient to lower overall suicide rates. During the same period, there was a shift from less lethal methods, such as self-poisoning, to more lethal methods, such as hanging [30, 33, 34, 57, 61].

The increasing preference for highly lethal methods, such as hanging, reflects a stronger intention to commit suicide and raises significant public health concerns [30, 33, 34, 57, 61]. This is because effective preventive measures for suicide by hanging are difficult to implement. This method is readily accessible and can only be effectively prevented in institutional settings, such as hospitals and prisons [34].

In this context, Brazil observed an 89.45% increase in suicides among women by hanging and strangulation/suffocation, comparing the proportional mortality rates between the periods 1980–1989 and 2010–2019 (28.29% vs. 53.58%). Conversely, suicides by firearms decreased by 66.72% (12.14% vs. 4.04%), and suicides by self-poisoning decreased by 9.45% (26.43% vs. 23.93%). Similar trends were observed across all regions of the country [32, 33].

We believe that the economic crisis faced by Brazil from 2005 to 2019 may have contributed to the increased risk of suicide by autointoxication, hanging, and strangulation/suffocation among women in Brazilian regions [32, 33, 57, 58, 90–94]. The relationship between economic crisis and suicide has been widely studied, starting with Durkheim in the 19th century. According to Durkheim (2019), economic crises create imbalances and disturbances in the collective order, reducing integration and social control, and thereby increasing the risk of suicide [7]. In line with this perspective, the theoretical model developed by Case and Deaton (2015) explores how economic stagnation is associated with rising suicide rates, a phenomenon they term 'deaths of despair' [9]. During periods of economic crisis and fiscal austerity, there is an observed increase in 'diseases of despair' (such as illicit drug use, alcohol abuse, and suicide attempt) and 'deaths of despair' (including suicide, overdose, and liver cirrhosis) [7].

Brazil, between 1980 and 2019, experienced several economic crises that deeply shaped the country's social and economic landscape [92–95]. The prolonged crisis of the 1980s, characterized by high inflation and frequent changes in economic plans and currency, was followed by a recession in the 1990s, despite efforts to stabilize the economy with the Real Plan. In the early 2000s, the country experienced a period of growth, but soon faced a new recession due to the global economic crisis of 2008. From 2014 onwards, Brazil entered a phase of economic decline with a reduction in GDP for fifteen quarters, exacerbated by fiscal adjustment measures that reduced spending on health and education and imposed labor and pension reforms [92–95].

These economic changes had repercussions on deaths of despair and suicide rates in the Brazilian population [95, 96]. Between 2003 and 2018, suicides accounted for 85% of deaths of despair, with a notable increase in suicide rates following the economic crisis of 2014 [95]. This pattern is supported by analyses showing a progressive rise in monthly suicide rates from 2011 to 2017, reflecting a direct response to the economic crisis [96].

In theory, economic crises would equally affect all age groups; however, studies indicate that these events have a more pronounced impact on the economically active population [32, 33, 49, 69, 85, 87, 94, 96, 97], resulting in a cohort effect [35, 36]. The cohort effect describes how prolonged exposure to social, economic, cultural, political, and health phenomena can affect different generations in differentiated ways [35, 36].

Since the 1950s, Brazilian society has undergone significant social, cultural, economic, and demographic transformations. Among these changes are rapid and unplanned urbanization, accelerated shifts in the population's age structure, alterations in family composition, such as the growth of single-parent families and those headed by women, and the increasing presence of women in the labor market [98, 99]. These changes occurred within the context of turbulent economic cycles, marked by several crises [92–95]. Such transformations have contributed to a reduction in social solidarity and cohesion, creating anomic environments that favor increases in crime, homicides, and suicidal behaviors [7, 100, 101].

In this regard, our study reveals variations in the risk of suicide among women based on the method used and the cohort. Women from more recent cohorts (1955 to 2009) showed a reduced risk of suicide by firearms and poisoning, while there was an increase in the risk of suicide by hanging, strangulation, and suffocation among these generations. This pattern suggests a substitution of suicide methods across generations, reflecting changes in the availability and attitudes toward these methods.

The higher risk of suicide by hanging, strangulation, and suffocation among women from younger cohorts may correlate with greater awareness of lethality and availability due to increased access to the internet and social media [69]. Additionally, women from Generation X (1965 to 1980) and Generation Y (1981 to 1996) are part of large cohorts and may experience the phenomenon known as cohort discontinuity [99–101]. Cohort discontinuity refers to the difficulties that members of large cohorts will face throughout their life cycle in accessing quality education, entering the labor market, and achieving economic security [99–101]. According to Pampbell (1996), cohorts experiencing this phenomenon are more likely to suffer from social isolation and anomie, increasing the risk of involvement in violent behaviors, including self-directed violence [101].

In addition to the impact of cohort discontinuity, younger Brazilian women also faced a long period of exposure to the sociocultural changes of the 1960s and 1970s, which challenged patriarchy and traditional gender norms [13, 15, 16]. Despite the achievements of women in recent decades, such as entry into the labor market, the approval of divorce, and ascension to high management and political positions, Brazil still maintains a conservative patriarchal culture. This culture perpetuates significant gender inequalities, particularly affecting women [13, 15, 16, 38].

In 2022, Brazilian women continued to have lower earnings and higher rates of unemployment and underemployment compared to men [85]. Additionally, they are primarily responsible for unpaid caregiving for children, household chores, and partners, resulting in a double or triple workload. This burden can lead to emotional stress, increasing the risk of mental disorders, suicidal behaviors, and suicides [10, 16, 18, 23].

The higher risk of suicide using highly lethal methods, such as hanging, among younger generations of Brazilian women suggests a more determined intention to consummate the act, reflecting a situation of extreme pain and suffering [6, 12, 19–21]. Although all women are vulnerable to gender-based violence, this violence is particularly prevalent among the younger generations, especially among women aged 15 to 49 [6, 12, 19–21]. By challenging traditional gender roles, these women increase their risk of facing violence as they potentially challenge male supremacy. This supremacy is, in turn, legitimized by society, which resorts to violence—even fatal violence—to preserve its status [15–18].

In the 2000s, thanks to the efforts of women's movements and the feminist movement, several legal measures against violence against women were implemented in Brazil, such as the Maria da Penha Law (2006) and the Femicide Law (2015). These advances expanded the debate on violence against women, which gained prominence in society. However, these movements for social structural reform have been accompanied by conservative counter-reform movements that advocate for the maintenance of traditional gender roles. Among these conservative movements, religious fundamentalism has waged a war against women's rights and the LGBTQI+ community [15–18].

In this context, the Brazilian state, through its institutions, has been complicit in the violence suffered by women by maintaining an inadequate budget for prevention, protection, and social assistance policies for women facing violence. Twelve years after the implementation of the Maria da Penha Law, only 5.2% of Brazil's 5,572 municipalities had shelters, only 10% of cities had specialized services for women who are victims of sexual violence, and only 8.3% of cities had specialized police stations for women [25]. This situation has contributed to an increased risk of suicide among women with a history of domestic violence. In Brazil, women who reported violence had a suicide rate 30 times higher than women without a history of violence [30].

The results of this study highlight the need for Brazil to adopt preventive measures for suicides among women involving hanging, poisoning, and firearms. These measures should consider regional particularities, including identity formation, ethnic-racial composition, and the structure of the psychosocial support network [59, 60, 102]. Additionally, it is important to consider the differences in suicide risk between the methods studied, according to age and cohort.

The increase in the risk of suicide by poisoning among younger age groups from 2010 to 2014, and among younger cohorts in the Northeast region, indicates the need for effective implementation of the National Policy on Self-Harm and Suicide Prevention (2019) [102]. This policy advocates for controlling the sale of pesticides, household chemicals, and firearms (Brazil, 2019). These are essential actions, as the availability of firearms, insecticides, and pesticides facilitates impulsive suicides that do not require detailed planning or deep technical knowledge [86–90]. In cases of suicide by hanging, younger generations have shown a higher likelihood of using this method, indicating intense suffering and a high intent to die [86–90]. For these generations, teaching coping strategies to deal with stressful situations has proven to be an effective measure [86–90].

Reducing access to lethal means of suicide is an important strategy, but on its own, it is not sufficient to lower suicide rates, as methods can be substituted, especially in individuals with psychiatric disorders who are more impulsive [86, 90]. Therefore, intersectoral actions involving family, community, and health and education professionals are needed to identify young people in suicidal crisis and refer them, in a timely manner, to psychosocial assistance.

## Limitations and strengths of the study

The poor certification and underreporting of death records in the Mortality Information System (SIM) are limitations of this study. However, we used demographic and epidemiological techniques to mitigate these issues and produce more reliable mortality rates.

Another limitation pertains to the Age-Period-Cohort (APC) models, as there is no consensus in the literature on the best method to address the identification problem that arises from the linear relationship between age, period, and cohort. This relationship allows for the generation of infinite maximum likelihood models with different parameters and estimates that produce the same predictions for any combination of these factors, thereby hindering the

estimation of the full model [35, 36]. In this study, we adopted the methodology most frequently recommended by authors who have compared classical statistical methods: estimable functions [35, 36]. Additionally, the high risk of suicide by hanging, strangulation, and suffocation in recent generations should be interpreted with caution, as it may be related to the smaller number of observations in these cohorts. Nonetheless, we chose to analyze suicides in the 10 to 19 age group due to the upward trend in suicides observed in this group in Brazil [29, 30].

The main contribution of this study was to estimate corrected suicide rates to account for poor certification and record coverage, according to the primary methods of suicide used over the past four decades. It also estimated Age-Period-Cohort (APC) models by method and geographical region. These results can aid in the assessment and planning of suicide prevention and control measures for women, differentiated by method and risk according to age and cohort.

## Conclusion

The findings of our study revealed a shift in suicide methods among Brazilian women, with a decreased risk of suicide by firearm and an increased risk of suicide by self-poisoning (from 2005 to 2014) and hanging (from 2005 to 2015). Additionally, we identified variations in age and cohort effects according to the method used. Older women exhibited higher rates of suicide by hanging, while younger women showed a higher risk of suicide by firearm and autointoxication. Regarding cohort effects, older generations had a higher risk of suicide by firearm and autointoxication. In contrast, younger generations of women had a higher risk of suicide by hanging. Our findings underscore the necessity for targeted and integrated suicide prevention strategies, taking into account the distinct experiences and contexts faced by women across different generations and age groups.

## Supporting information

**S1 Table. GATHER checklist of information that should be included in reports of global health estimates.**
(DOCX)

**S2 Table. Stages of the process for correcting suicides by method of perpetration for information quality and coverage of death records.**
(DOCX)

**S3 Table. Suicide rates per 100,000 women, by means of perpetration, stages of correction of records, decade, and region.** Brazil, 1980–2019.
(DOCX)

**S4 Table. Deviance, Akaike Information Criterion (AIC) an p-value analysis in sequential construction of age, period, and cohort models for suicides in women by methods, according to Brazil its major regions, from 1980 to 2019.**
(DOCX)

**S5 Table. Relative risk and 95% confidence interval for the effect of period and cohort for suicides in women, according.**
(DOCX)

## Author Contributions

**Conceptualization:** Karina Cardoso Meira, Raphael Mendonça Guimarães, Eder Samuel Oliveira Dantas.

**Data curation:** Karina Cardoso Meira, Eder Samuel Oliveira Dantas.

**Formal analysis:** Karina Cardoso Meira, Raphael Mendonça Guimarães, Glauber Weder Santos Silva, Rafael Tavares Jomar.

**Investigation:** Karina Cardoso Meira, Glauber Weder Santos Silva.

**Methodology:** Karina Cardoso Meira, Raphael Mendonça Guimarães, Rafael Tavares Jomar, Eder Samuel Oliveira Dantas.

**Project administration:** Karina Cardoso Meira.

**Resources:** Karina Cardoso Meira.

**Software:** Karina Cardoso Meira.

**Supervision:** Karina Cardoso Meira, Eder Samuel Oliveira Dantas.

**Validation:** Raphael Mendonça Guimarães, Glauber Weder Santos Silva, Rafael Tavares Jomar, Eder Samuel Oliveira Dantas.

**Visualization:** Karina Cardoso Meira, Raphael Mendonça Guimarães, Glauber Weder Santos Silva, Rafael Tavares Jomar, Eder Samuel Oliveira Dantas.

**Writing – original draft:** Karina Cardoso Meira, Raphael Mendonça Guimarães, Glauber Weder Santos Silva, Rafael Tavares Jomar, Eder Samuel Oliveira Dantas.

**Writing – review & editing:** Karina Cardoso Meira, Raphael Mendonça Guimarães, Glauber Weder Santos Silva, Rafael Tavares Jomar, Eder Samuel Oliveira Dantas.

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
