## [Decision Letter · Decision Letter 0]

7 May 2024

PONE-D-24-08405Suicide Methods Among Brazilian Women from 1980 to 2019: Influence of Age, Period, and Cohort

PLOS ONE

Dear Dr. Meira,

Thank you for submitting your manuscript to PLOS ONE. After careful consideration, we feel that it has merit but does not fully meet PLOS ONE’s publication criteria as it currently stands. Therefore, we invite you to submit a revised version of the manuscript that addresses the points raised during the review process.

**ACADEMIC EDITOR: **

I regret to inform you that we are unable to accept your manuscript for publication in its current form. The reviewers have provided valuable feedback on your manuscript, with two out of three reviews being positive. 

However, the negative review raises significant concerns about the research methodology and the results. 

Therefore, we recommend that you thoroughly address all the comments provided by the reviewers and revise your manuscript accordingly.

Please pay close attention to the reviews It is essential that you address these concerns comprehensively in your revised manuscript to ensure its suitability for publication. We understand that revising your manuscript may require substantial effort, but we believe that addressing the reviewers' comments will significantly improve the quality and impact of your research. We encourage you to carefully consider all the feedback provided and to make the necessary revisions accordingly.

We look forward to receiving your revised manuscript.

Kind regards,

Claudio Alberto Dávila-Cervantes, Ph.D.

Academic Editor

PLOS ONE

Reviewers' comments:

Reviewer's Responses to Questions

**Comments to the Author**

1. Is the manuscript technically sound, and do the data support the conclusions?

Reviewer #1: Yes

Reviewer #2: Partly

Reviewer #3: Yes

2. Has the statistical analysis been performed appropriately and rigorously? 

Reviewer #1: Yes

Reviewer #2: No

Reviewer #3: Yes

3. Have the authors made all data underlying the findings in their manuscript fully available?

Reviewer #1: Yes

Reviewer #2: Yes

Reviewer #3: No

4. Is the manuscript presented in an intelligible fashion and written in standard English?

Reviewer #1: Yes

Reviewer #2: Yes

Reviewer #3: Yes

5. Review Comments to the Author

Reviewer #1: The manuscript examines age, period, and cohort effects on suicide by methods used among women in Brazil.

I find the paper well-written; it has the potential to contribute to the existing literature on the topic. However, I have some concerns I would like the authors to consider.

Introduction

1.) “These ideals are constantly conveyed to individuals through gender technologies performed by family, religion, school, art, media, among others [6,10,13,19]. Zanello [19] asserts that men's subjectivity is configured by gender technologies that emphasize the devices of sexual and economic efficiency, while women's subjectivity is shaped by love and maternal devices.”

What do the authors consider as relevant gender technologies in this context? Please consider clarifying by using some examples.

2.) “…mortality in women according ccording to the most common means of perpetration…”

Please correct this sentence.

Materials and methods

3.) The authors have included in their study women aged ten years and older. While it is quite a frequent practice to combine data on adults and adolescents when examining suicide, there is ample evidence showing significant differences in the suicide behavior of adolescents compared to adults. Why did the authors decide to include the population aged 10–19? Taking into account the potentially different etiology behind suicide in adolescents, please consider excluding these age groups or justifying in the text the decision to include them. In addition, among the limitations, the authors mentioned fewer observations in younger generations.

4.) Table 1: “W75 e W76”

Please correct the typographical error.

5.) “Here, i = 1, ..., I; j = 1, ... J; k = 1, ..., K; and where K = I + J-1.”

Since lowercase letters i, j, and k are only used in the equation presented later in the text, this sentence might make more sense after introducing Holford's proposal.

6.) “… the effect of period …”

Unlike in the main text, this is in the equation.

7.) “cohort (k = 1, . . . , k = I + J − 1 = 22)”

Here, the second lowercase k in the brackets seems to be a typographical error.

Results

8.) All results were shown after correcting the death records. Did the authors assess the trends – not the death rates – using the data before correction? Were there any notable differences between trends based on corrected versus uncorrected records (including potential regional differences)?

Discussion

9.) “In developing countries, the XX (1965-1979), late Y (1973-1980), and Millennial (1981-1991) generations are experiencing cohort discontinuity.”

Usually, in the literature, Generation X refers to people born between 1965 and 1980, while Generation Y – also known as Millennials – refers to those born from the early 1980s to mid-1990s. Therefore, this sentence might be somewhat confusing.

10.) “…a pattern that was maintained when estimating the APC models. drift trend…”

Please correct the typographical error.

11.) “During the study period, Brazil went through several economic crises [72-74,82]…”

Please consider explaining a bit more about the mentioned economic crises, especially about the specific timing of these events.

Supporting Information

12.) S1. Table

Please correct the typographical errors, such as:

“Lon-term consequences…”

“(Stage 6):- ”

“aP roportion of…”

“CID-9 and CID-10 Codes”

Reviewer #2: ID: PONE-D-24-08405

Title: Suicide Methods Among Brazilian Women from 1980 to 2019: Influence of Age, Period, and Cohort

Thank you for providing a chance to review this manuscript.

Detailed information:

Abstract

Overall: If the journal does not have special requirements, I recommend presenting the abstract section in order of background/objective, methods, results, conclusions.

Line 25-27, Page 2: What is the meaning of this sentence placed here? Also, please add the research purpose.

Line 31, Page 2: What is the purpose of using model estimation? Please provide a detailed description of the research design in the methodology section and what data indicators were collected. What is the sample size?

Line 31-45, Page 2: 1) Is this the results section? There are no transitional words. Please pay attention to the division of each section in the abstract to help readers clarify. 2) Please provide the numerical values of key indicators as evidence support, not just language expression. 3) Display the main results. Reduce excessive elaboration.

Please explain the conclusion in one to two sentences.

Keywords: There are no keywords in the text you provided, please pay attention to the details.

Introduction

Line 52, page 3: Complex? Where is it reflected? The first paragraph only shows the high incidence of suicidal behavior. Please modify and improve. Also, this is a new paragraph, what does "it" refer to? This statement does not seem to have a causal relationship with the preceding text.

Line 53-54, Page 3: What is the significance of emphasizing the different impacts on different genders? Where are the different impacts reflected? This is abrupt. Please modify.

Line 58-60, Page 3: Please use appropriate transitional sentences to continue the two paragraphs.

Line 86-87, Page 4: Please elaborate more on "crucial". And clarify the significance of conducting this research.

Line 92-97, Page 5: The introduction of the keywords "age period cohort effect" or "age period cohort" in this paragraph is too concise. Please provide more explanation to assist in explaining the significance of the research.

Line 102, Page 5: Please propose reasonable research hypotheses based on your research objectives.

Overall: Overall, the logic of the preface needs to be strengthened, and some keywords are not even elaborated in the preface section. 1) As you mentioned in your keyword about mental health, it was not elaborated on in the introduction section. 2) Previously, it was often mentioned that "the first two decades of the 1980s". Is there any difference or connection between the data from this period and the data from your research period?

Materials and Methods

Study design and data sources

Line 107, Page 5: Please provide a detailed explanation of which recommendations.

The research design is too brief to fully understand your research ideas and strive for perfection.

Study variables:

I don't quite understand. The title is called Study Variables, but the content does not clearly summarize all the research variables and introduce the collection of key variables. Please indicate which variables and indicators were collected for this study.

Table 1: A three-line table is recommended if not specifically required by the journal.

Results

1) Tabulate and summarize the demographics of the study population, if available in the database, such as the age distribution.

2) I think the use of a heat map to depict the distribution of different suicide methods among women would be a much clearer and more intuitive way to visualize the results of the study.

3) All the figures in the text seem like they could be further optimized to make them more aesthetically pleasing.

4) The descriptions of the results in each section seem somewhat confusing and lengthy, please summarize the findings in summary terms.

Discussion

Line 370-385, page 18-19: The discussion in this article should be addressed to women in the Brazilian region. Is it possible to find more theories to support the results of this study?

Overall: 1) Please integrate and summarize some of the passages according to content and logic. 2) Based on the findings of this study, what actions or measures can be taken by the relevant authorities or the government to improve the situation of female suicide?

I'm sorry to say that I think this article only shows descriptive results of the impact of APC on female suicide in Brazil, and does not reflect the actual value and further improvements.

Thank you and my best,

Your reviewer

Reviewer #3: Dear authors,

I really enjoy reading your paper. Addressing changes in suicide methods adopted by women over the time and the regional differences are really important to implement public policies to fight this public health problem. However, bellow I include some some suggestions to improve this manuscript.

Abstract

I suggest the authors go direct to the goal of the paper.

The results about age differences in suicide rates should be included.

Introduction:

I suggest the authors excluding lines 92-102 since the subjects were addressed in paragraphs above.

I didn’t see nothing about policies in fighting gender inequities or promoting women health in Brazil in the introduction and in discussion. It is essential to this paper to address this theme.

Results

OK

Methods

I only suggest you to include your data sheet in a repository where others can have free access.

Discussion

Discussion should be more focused on your main results. In lines 236-237 you brought your main goal. So, please, be focused on it. Additionally, you should brought these results to the first paragraph of the discussion.

In addition, the discussion is confused and needs to be reorganized. First discuss all results about age, methods of suicide and, then, cohort.

Are there no studies in Latin America in the same subject? Why to compare results with other countries with greater sociocultural and economic differences?

I didn’t see the point to the paragraph in lines 441-445.

No need to mention COVID-19 here.

Where is the limitation of this study?

Again, I suggest you discuss about public policies fighting gender inequities in Brazil as possible regional barriers to the policies implementations.

Conclusion

It needs to be rebuild answering the main goal of your study.

Minor point:

Please, the text should be review by native English speakers.

6. PLOS authors have the option to publish the peer review history of their article (what does this mean?). If published, this will include your full peer review and any attached files.

Reviewer #1: No

Reviewer #2: No

Reviewer #3: No

---

## [Author Response · Author response to Decision Letter 0]

6 Aug 2024

PONE-D-24-08405

Suicide Methods Among Brazilian Women from 1980 to 2019: Influence of Age, Period, and Cohort

PLOS ONE

Dear Dr. Meira,

Thank you for submitting your manuscript to PLOS ONE. After careful consideration, we feel that it has merit but does not fully meet PLOS ONE’s publication criteria as it currently stands. Therefore, we invite you to submit a revised version of the manuscript that addresses the points raised during the review process.

ACADEMIC EDITOR:

I regret to inform you that we are unable to accept your manuscript for publication in its current form. The reviewers have provided valuable feedback on your manuscript, with two out of three reviews being positive. 

However, the negative review raises significant concerns about the research methodology and the results. 

Therefore, we recommend that you thoroughly address all the comments provided by the reviewers and revise your manuscript accordingly.

Cite all versions? You can cite all versions by using the DOI 10.5281/zenodo.13172957.

Response: 

Dear Editor,

We extend our gratitude for the excellent contributions of the reviewers and for the opportunity to revise our manuscript. In the Revised Manuscript with track changes, sentences highlighted in yellow have been removed or replaced by sentences in green. Some sentences that were reordered have been marked in blue. We have organized the responses to the reviewers by topics according to the following order: abstract, introduction, methodology, results, discussion, and conclusion.

Thank you for your attention to these revisions. We look forward to your feedback.

Sincerely,

The authors.

3. Have the authors made all data underlying the findings in their manuscript fully available?

 Abstract

Reviewer #2

Overall: If the journal does not have special requirements, I recommend presenting the abstract section in order of background/objective, methods, results, conclusions.

 Response:We would like to thank the reviewer for their valuable contributions, following the journal's guidelines (https://journals.plos.org/plosone/s/submission-guidelines), we made the changes to the abstract. We made an effort to follow the suggestions of reviewers 2 and 3 and maintain the number of words determined by the Journal. Thus, we begin the abstract from the objectives, in the methodology we detail the methodology used in the APC analyses In the results we present the total number of suicides evaluated during the study period, and we explain to readers what was considered statistically significant.

Abstract 

Objective: To analyze the effect of age, period, and cohort on suicides among women by hanging, strangulation, suffocation, firearms, and autointoxication in different Brazilian regions from 1980 to 2019.Methods: Ecological time-trend study employing estimable functions to estimate APC models, facilitated through the Epi library of the R statistical program, version 4.2.1. Specific rates by age group per 100,00 women and relative risks by period and cohort were estimated using this method. Results: Between 1980 and 2019, 49,997 suicides among women were reported using the methods under study. Higher suicide rates per 100,000 women were observed in the South using strangulation and suffocation (2.42), while lower firearm suicide rates were observed in the Northeast (0.13). After adjusting the APC model, there was an increase in age-specific rates with advancing age across all regions for suicides by hanging, strangulation, and suffocation. In contrast, suicides by firearms and autointoxication showed a decrease in rates with advancing age. The period effect indicated an increased risk of suicides by hanging, strangulation (RR >1 and p<0.05) in the five-year intervals of the 2000s in the North, Southeast, and South regions. During the same period, there was an increased risk of suicides by autointoxication in the Southeast, South, and Northeast (RR>1, p<0.05). Suicides by firearms exhibited a statistically significant reduction in the risk of death from 2005 to 2019 in the Southeast and South regions, and from 2005 to 2014 in the Northeast and Midwest. The observed increase in the North region was not statistically significant (RR>1, p>0.05). The cohort effect demonstrated an increased risk of suicides by hanging, strangulation in younger cohorts (RR>1, p<0.05), whereas other methods showed an elevated risk in older cohorts relative to the 1950-1954 generation. Conclusion: The results presented here may suggest changes in suicide method preferences between 1980 and 2019.

Key-words: Suicide, women,Gender and Health; Mental Health; Age-period-cohort effect

 Line 25-27, Page 2: What is the meaning of this sentence placed here? Also, please add the research purpose.

Response: We restructured the abstract, removed this sentence and inserted the objective of the study.

Objective: To analyze the effect of age, period, and cohort on suicides among women by hanging, strangulation, suffocation, firearms, and self-poisoning in different Brazilian regions from 1980 to 2019.

 Line 31, Page 2: What is the purpose of using model estimation? Please provide a detailed description of the research design in the methodology section and what data indicators were collected. What is the sample size?

Response: We restructured the methodology as requested.

Methods: Ecological time-trend study employing estimable functions to estimate APC models, facilitated through the Epi library of the R statistical program, version 4.2.1. Specific rates by age group per 100,00 women and relative risks by period and cohort were estimated using this method. 

 Line 31-45, Page 2: 1) Is this the results section? There are no transitional words. Please pay attention to the division of each section in the abstract to help readers clarify. 2) Please provide the numerical values of key indicators as evidence support, not just language expression. 3) Display the main results. Reduce excessive elaboration.

Response: We restructured the results as requested.

Results: Between 1980 and 2019, 49,997 suicides among women were reported using the methods under study. Higher suicide rates 100,000 women were observed in the South using strangulation and suffocation (2.42), while lower firearm suicide rates were observed in the Northeast (0.13). After adjusting the APC model, there was an increase in age-specific rates with advancing age across all regions for suicides by hanging, strangulation, and suffocation. In contrast, suicides by firearms and self-poisoning showed a decrease in rates with advancing age. The period effect indicated an increased risk of death (RR >1 and p<0.05) in the five-year intervals of the 2000s in the North, Southeast, and South regions. During the same period, there was an increased risk of suicides by self-poisoning in the Southeast, South, and Northeast (RR>1, p<0.05) and a reduction in the North (RR<1, p<0.05). Suicides by firearms exhibited a statistically significant reduction in the risk of death from 2005 to 2019 in the Southeast and South regions, and from 2005 to 2014 in the Northeast and Midwest. The observed increase in the North region was not statistically significant (RR>1, p>0.05). The cohort effect demonstrated an increased risk in younger cohorts (RR>1, p<0.05), whereas other methods showed an elevated risk in older cohorts relative to the 1950-1954 generation.

Reviewer #3

 I suggest the authors go direct to the goal of the paper.

 The results about age differences in suicide rates should be included.

Response: We thank you for your valuable contributions and have made the changes highlighted in red in the revised abstract, presented below.

Abstract 

Objective: To analyze the effect of age, period, and cohort on suicides among women by hanging, strangulation, suffocation, firearms, and autointoxication in different Brazilian regions from 1980 to 2019.Methods: Ecological time-trend study employing estimable functions to estimate APC models, facilitated through the Epi library of the R statistical program, version 4.2.1. Specific rates by age group per 100,00 women and relative risks by period and cohort were estimated using this method. Results: Between 1980 and 2019, 49,997 suicides among women were reported using the methods under study. Higher suicide rates per 100,000 women were observed in the South using strangulation and suffocation (2.42), while lower firearm suicide rates were observed in the Northeast (0.13). After adjusting the APC model, there was an increase in age-specific rates with advancing age across all regions for suicides by hanging, strangulation, and suffocation. In contrast, suicides by firearms and autointoxication showed a decrease in rates with advancing age. The period effect indicated an increased risk of suicides by hanging, strangulation (RR >1 and p<0.05) in the five-year intervals of the 2000s in the North, Southeast, and South regions. During the same period, there was an increased risk of suicides by autointoxication in the Southeast, South, and Northeast (RR>1, p<0.05). Suicides by firearms exhibited a statistically significant reduction in the risk of death from 2005 to 2019 in the Southeast and South regions, and from 2005 to 2014 in the Northeast and Midwest. The observed increase in the North region was not statistically significant (RR>1, p>0.05). The cohort effect demonstrated an increased risk of suicides by hanging, strangulation in younger cohorts (RR>1, p<0.05), whereas other methods showed an elevated risk in older cohorts relative to the 1950-1954 generation. Conclusion: The results presented here may suggest changes in suicide method preferences between 1980 and 2019.

Key-words: Suicide, women,Gender and Health; Mental Health; Age-period-cohort effect

B)Introduction

Reviewer #1: The manuscript examines age, period, and cohort effects on suicide by methods used among women in Brazil.

I find the paper well-written; it has the potential to contribute to the existing literature on the topic. However, I have some concerns I would like the authors to consider.

 “These ideals are constantly conveyed to individuals through gender technologies performed by family, religion, school, art, media, among others [6,10,13,19]. Zanello [19] asserts that men's subjectivity is configured by gender technologies that emphasize the devices of sexual and economic efficiency, while women's subjectivity is shaped by love and maternal devices.”

What do the authors consider as relevant gender technologies in this context? Please consider clarifying by using some examples.

Response: Thank you for the suggestion, we have inserted what was requested in the text, as shown in the text below.

The gender technologies are represented by linguistic codes and cultural representations such as cinema, media, games, and toys. These technologies are constantly performed by family, religion, school, art, media, and other institutions [6,10,13,19]. Zanello [19] asserts that men's subjectivity is configured by gender technologies that valorize the devices of sexual and economic efficacy, while women's subjectivity is shaped by the amorous and maternal devices. Thus, suicidal behaviors in men and women would be aligned with culturally defined gender scripts [20-23].

 “…mortality in women according ccording to the most common means of perpetration…”

Please correct this sentence

Response: Correction made

Thus, the aim was to address the following research question: Are there differences in the effects of age, period, and cohort on suicide mortality in women according to the most common means of perpetration between and within Brazilian regions?

Reviewer #2

 Overall: Overall, the logic of the preface needs to be strengthened, and some keywords are not even elaborated in the preface section.

Response: We appreciate the reviewer's valuable suggestions, based on them and the suggestions of other reviewers, we made structural changes to this section of the manuscript. We discussed the issue of mental health in Brazil, not with regard to the National Suicide Prevention Policy, in addition, we expanded the definition of the effect of age, period and cohort.

 Previously, it was often mentioned that "the first two decades of the 1980s". Is there any difference or connection between the data from this period and the data from your research period? 

Response: We believe the reviewer is referring to the one marked in red in the paragraph below. This period was included in our study, because we are evaluating the period from 1980 to 2019, the period from 2000 to 2019, including the first two decades of the 21st century. To make it clearer to readers, we decided to make a change in the text.

 Line 52, page 3: Complex? Where is it reflected? The first paragraph only shows the high incidence of suicidal behavior. Please modify and improve. Also, this is a new paragraph, what does "it" refer to? This statement does not seem to have a causal relationship with the preceding text.

Line 53-54, Page 3: What is the significance of emphasizing the different impacts on different genders? Where are the different impacts reflected? This is abrupt. Please modify.

 Line 58-60, Page 3: Please use appropriate transitional sentences to continue the two paragraphs.

Response Based on the suggestions above, we rewrote the first three paragraphs of the introduction.

In the text

Suicide affects people from different backgrounds, ethnical groups, socio-economic status and geographical locations [1–6]. Suicide can be defined as any deliberated act by which an individual’s death results directly or indirectly from a self-inflicted injury or poisoning [1-6]. Considered a sociocultural phenomenon, it is attributed to the interaction of individual and contextual factors [1-6]. At the individual level, noteworthy factors include depression, bipolar affective disorder, schizophrenia, anxiety, prior history of suicide attempts, excessive alcohol and drug use, and philosophical and existential questions [7-11]. Contextual factors include economic and health crises, as well as societies characterized by fragility in social cohesion [7-11].

The fragility in social cohesion has been documented in patriarchal societies with high levels of inequality in gender, race, and class relations [7,12-16]. In these societies, the illness and death of women and men are influenced by inequalities in gender relations [14-16]. The construct of gender is an inherent dimension of societal life, subject to various definitions over time [15-19]. Joan Scott [17] argues that gender emerges from the relationship of submission and oppression of women by men, shaping the notions of being male and being female in these societies. In other words, it defines what the thoughts, feelings, and behaviors of women and men should be through the ideal notions of masculinity and femininity, which are presented to individuals via gender technologies [6,10,13,19]. 

The gender technologies are represented by linguistic codes and cultural representations such as cinema, media, games, and toys. These technologies are constantly performed by family, religion, school, art, media, and other institutions [6,10,13,19]. Zanello [19] asserts that men's subjectivity is configured by gender technologies that valorize the devices of sexual and economic efficacy, while women's subjectivity is shaped by the amorous and maternal devices. Thus, suicidal behaviors in men and women would be aligned with culturally defined gender scripts [20-23].

 Line 86-87, Page 4: Please elaborate more on "crucial". And clarify the significance of

---

## [Decision Letter · Decision Letter 1]

27 Aug 2024

Suicide Methods Among Brazilian Women from 1980 to 2019: Influence of Age, Period, and Cohort

PONE-D-24-08405R1

Dear Dr. Meira,

We’re pleased to inform you that your manuscript has been judged scientifically suitable for publication and will be formally accepted for publication once it meets all outstanding technical requirements.

Kind regards,

Claudio Alberto Dávila-Cervantes, Ph.D.

Academic Editor

PLOS ONE

Reviewers' comments:

Reviewer's Responses to Questions

**Comments to the Author**

1. If the authors have adequately addressed your comments raised in a previous round of review and you feel that this manuscript is now acceptable for publication, you may indicate that here to bypass the “Comments to the Author” section, enter your conflict of interest statement in the “Confidential to Editor” section, and submit your "Accept" recommendation.

Reviewer #1: (No Response)

Reviewer #3: All comments have been addressed

2. Is the manuscript technically sound, and do the data support the conclusions?

Reviewer #1: Yes

Reviewer #3: Yes

3. Has the statistical analysis been performed appropriately and rigorously? 

Reviewer #1: Yes

Reviewer #3: Yes

4. Have the authors made all data underlying the findings in their manuscript fully available?

Reviewer #1: Yes

Reviewer #3: Yes

5. Is the manuscript presented in an intelligible fashion and written in standard English?

Reviewer #1: Yes

Reviewer #3: Yes

6. Review Comments to the Author

Reviewer #1: Most of my comments have been addressed. The only minor concern is related to the following comment:

6.) “… the effect of period …”

Unlike in the main text, this is in the equation.

I believe this issue remained in the manuscript.

Reviewer #3: (No Response)

7. PLOS authors have the option to publish the peer review history of their article (what does this mean?). If published, this will include your full peer review and any attached files.

Reviewer #1: No

Reviewer #3: No

---

## [Editor Report · Acceptance letter]

20 Sep 2024

PONE-D-24-08405R1 

PLOS ONE

Dear Dr. Meira, 

I'm pleased to inform you that your manuscript has been deemed suitable for publication in PLOS ONE. Congratulations! Your manuscript is now being handed over to our production team.

Kind regards, 

on behalf of

Mr. Claudio Alberto Dávila-Cervantes 

Academic Editor

PLOS ONE